# Carbon mitigation potential afforded by rooftop photovoltaic in China

Zhixin Zhang [1,2,3], Min Chen [1,4,5,6] ✉, Teng Zhong [1,3,5], Rui Zhu[7], Zhen Qian [3], Fan Zhang[8], Yue Yang[3], Kai Zhang[3], Paolo Santi [9], Kaicun Wang[10], Yingxia Pu[2,5,11], Lixin Tian[12,13], Guonian Lü [1,3,5] ✉ & Jinyue Yan [14,15] ✉

Rooftop photovoltaics (RPVs) are crucial in achieving energy transition and climate goals, especially in cities with high building density and substantial energy consumption. Estimating RPV carbon mitigation potential at the city level of an entire large country is challenging given difficulties in assessing rooftop area. Here, using multi-source heterogeneous geospatial data and machine learning regression, we identify a total of 65,962 km$^2$ rooftop area in 2020 for 354 Chinese cities, which represents 4 billion tons of carbon mitigation under ideal assumptions. Considering urban land expansion and power mix transformation, the potential remains at 3-4 billion tons in 2030, when China plans to reach its carbon peak. However, most cities have exploited less than 1% of their potential. We provide analysis of geographical endowment to better support future practice. Our study provides critical insights for targeted RPV development in China and can serve as a foundation for similar work in other countries.

Countries around the world are cooperating extensively[1] to tackle the global challenge of climate change due to greenhouse gas emissions. Currently, as one of the top $CO_2$ emitters of the world, with annual carbon emissions in 2020 exceeding 10 billion tons[2], China has made a series of carbon mitigation efforts[3,4]. At the 75th session of the United Nations General Assembly, China committed to reach its carbon peak by 2030 and achieve carbon neutrality by 2060[5]. As a major energy producer with high fossil coal dependency, China's power sector accounts for approximately half of the country's energy-related carbon

emissions[6]. Solar photovoltaic systems have been recognized as a promising technology that can decarbonize the power sector[7], with an estimated potential to meet 25–49% of the global electricity demand by 2050[8]. In 2020, China's cumulative installed PV capacity reached 253 GW, accounting for one-third of the world's capacity[9]. Recently, distributed photovoltaic (DPV) systems are preferred, due to their broad applicability, ease of local implementation, lower peak demand, and fewer transmission problems[10]; the share of DPV systems installed nationwide rose from 13% in 2016 to 31% in 2020[11]. The very large

[1]Key Laboratory of Virtual Geographic Environment (Ministry of Education of PRC), Nanjing Normal University, Nanjing 210023, China. [2]School of Geography and Ocean Science, Nanjing University, Nanjing 210023, China. [3]School of Geography, Nanjing Normal University, Nanjing 210023, China. [4]International Research Center of Big Data for Sustainable Development Goals, Beijing 100094, China. [5]Jiangsu Center for Collaborative Innovation in Geographical Information Resource Development and Application, Nanjing 210023, China. [6]Jiangsu Provincial Key Laboratory for NSLSCS, School of Mathematical Science, Nanjing Normal University, Nanjing 210023, China. [7]Institute of High Performance Computing (IHPC), Agency for Science, Technology and Research (A*STAR), 1 Fusionopolis Way, Singapore 138632, Republic of Singapore. [8]Department of Civil and Environmental Engineering, The Hong Kong University of Science and Technology, Hong Kong, China. [9]Senseable City Laboratory, Department of Urban Studies and Planning, Massachusetts Institute of Technology, Cambridge, MA 02139, USA. [10]Sino-French Institute for Earth System Science, College of Urban and Environmental Sciences, Peking University, Beijing 100871, China. [11]Jiangsu Provincial Key Laboratory of Geographic Information Science and Technology, Nanjing 210023, China. [12]Research Institute of Carbon Neutralization Development, School of Mathematical Sciences, Jiangsu University, Zhenjiang 212013, China. [13]Key Laboratory for NSLSCS, Ministry of Education, School of Mathematical Sciences, Nanjing Normal University, Nanjing 210023, China. [14]Department of Building Environment and Energy Engineering, The Hong Kong Polytechnic University, Kowloon, Hong Kong, China. [15]Future Energy Center, Mälardalen University, Västerås 72123, Sweden. ✉e-mail: chenmin0902@njnu.edu.cn; gnlu@njnu.edu.cn; jjyan@polyu.edu.hk

building stock in China, which has experienced an urbanization and construction boom in recent decades[12], suggests favorable site conditions for the development of DPV systems[13]. Also, the integration of PV with existing infrastructure offers new features for the development and implementation of renewable distributed power generation[14,15]. To better understand the development potential of RPVs and the contributions of the system to carbon mitigation efforts, nationwide assessments are urgently needed.

Evaluating China's RPV carbon mitigation potential at the city level is critical for targeted sustainable energy planning[16,17]. A few PV application studies at the city level have been presented as case studies, with more in-depth analyses based on field investigation[18]. However, there are over 300 prefecture-level cities in China, each with different geographical conditions and socioeconomic characteristics. The existing studies of a single city are limited to local development and therefore cannot be used as adequate references for other cities in China[19]. Notably, RPV systems are deployed in a decentralized manner, which increases the complexity of their assessment[20], with the greatest inaccuracy attributed to the delineation and calculation of building rooftops[21]. In previous studies, detailed mapping data of rooftop areas were acquired via high-resolution remote sensing and at high information processing costs; therefore, this data is difficult to obtain for the majority of cities in China[22]. Most of the existing multicity studies apply easily accessible national statistics, such as floor area and land use, to indirectly determine the rooftop area[11,23,24]. These methods are effective when the data for rooftop area is not available; however, the accuracy of the assessments is limited by the quality of the input data[25]. In addition, national statistics are generally aggregated at the provincial or higher level, which leads to substantial gaps in city-level assessments[26,27]. Detailed geospatial data with higher accuracy is essential to estimate the RPV potential. Therefore, there is a need to develop new methods to address the lack of data for the total rooftop area on a national scale.

Currently, urban construction in China is in a phase of rapid development. Studies on urban information acquisition are constantly emerging and being improved with the support of new technologies such as artificial intelligence, which provides new opportunities for the establishment of refined urban measurements[28–31]. To address the main data and computational bottleneck generally experienced in city-level estimations, an accurate method of obtaining rooftop areas at the scale of an entire large country is crucial. For this purpose, we introduce an accurate machine learning-based regression analysis that relied on multisource heterogeneous geospatial data, in particular a vectorized rooftop area dataset, which was developed by us previously and covered 16% of the total area of China[32]. Our study quantifies the RPV carbon mitigation potential of 354 Chinese cities, covering 88% of the total area of the country in 2020. In addition, we clarify the geographical heterogeneity in the RPV carbon mitigation potential and reveal the reasons for this variation through clustering analysis. Considering different scenarios of urban land expansion and power mix transformation, we estimate the future changes of the potential in 2030, when China plans to reach its carbon peak. Additionally, the current status of PV installation, energy consumption, and grid emissions are introduced to support the sustainable exploitation of the estimated potential.

## Results

### Study area and city representations
This study refines building rooftop area measurements from a large-scale vector building rooftop area dataset that we created in a previous work, the overall accuracy and F1 score of which are 98% and 83%, respectively[32]. The dataset covers 16% of the total area of China, containing cities with different administrative levels and geographical locations that are economically, politically, and geographically diverse. Based on this dataset, the national rooftop area was obtained using a machine learning-based regression model. The conversion of rooftop area to solar potential was carried out using a surface solar radiation dataset for China with a high-resolution (10 km), which performed better than most conventional products[33]. To assess the carbon mitigation of the RPVs, we used a carbon mitigation factor, which measures the carbon mitigation when RPV systems replace the electricity generated by existing and newly-built power plants in the local power grid. The carbon mitigation factor was determined through the baseline emission factors of China's regional power grid[34] (Supplementary Notes 1–2 and Supplementary Table 1).

As the baseline emission factors of Hong Kong, Macau, Taiwan, and Tibet were not available, the study area of this work included 354 Chinese cities, covering an area of over 8 million km$^2$ (Supplementary Note 3). Among them, the rooftop areas of 86 cities were obtained based on the vectorized dataset, while those of the remaining 268 cities were extrapolated by regression analysis through multisource heterogeneous geospatial data and machine learning (Fig. 1). In this study, the RPV carbon mitigation potential was defined as the $CO_2$ mitigation resulting from the replacement of grid electricity by electricity generated by RPV systems. The results were based on assumptions of rooftop availability of 35%, PV panel conversion efficiency of 20%, and overall RPV system efficiency of 80%. The main input and output data for city-level assessments are provided in Supplementary Data 1. To explain the variations due to different RPV system parameters and rooftop availability, we recorded a set of national RPV carbon mitigation potentials in 2020 for different settings (Supplementary Table 2). Additional assumptions and limitations in the interpretation of the main results are documented in the Methods and Discussion sections, respectively.

### Extrapolation and validation of rooftop area
The extrapolation of the rooftop area was performed with cells as the basic unit; each cell had an area of 10 km$^2$. The study area was partitioned into 1,045,022 square cells. The 86 cities having available measured rooftop area values were selected as the sample area, with a total of 191,370 cells (Supplementary Table 3). To build an extrapolation regression model of the rooftop area, four indicators closely related to urban construction, including road length, built-up area, population size, and night light intensity were selected as explanatory variables. Publicly accessible and high-quality data including vector road network data from OpenStreetMap (OSM)[35], land cover data from the Environmental Systems Research Institute (ESRI) (resolution of 10 m)[36], raster population data from WorldPop (resolution of 100 m)[37], and a nighttime light map from Earth Observation Group (resolution of 500 m)[38] were used in this study (Fig. 2).

The selected explanatory variables were found to correlate well with the rooftop area (Fig. 3). Sensitivity analysis was also conducted to prove the effectiveness of the selected variables (Supplementary Note 4 and Supplementary Tables 4–6). We applied the random forest algorithm to build a regression model based on the divided data. The algorithm's hyperparameters were adjusted by 10-fold cross-validation (Supplementary Table 7), and finally to build a model that could predict the rooftop area accurately. In our tests, the regression model based on the random forest algorithm provided better accuracy (goodness of fit, $R^2$) and less overall error (mean absolute error, MAE) when compared with other models (Supplementary Note 5 and Supplementary Table 8).

The trained model was applied to the non-sampled areas in 268 cities to obtain the rooftop area prediction values. To validate the extrapolation results, three representative cities were selected in each of the six geographic regions of China to form a validation dataset containing 18 cities and covering 134,29 cells. To obtain ground truth data for validation, we used the deep learning semantic segmentation method (applied in a previous work[32]) to extract the building rooftops from high-resolution satellite images (Supplementary Note 6 and

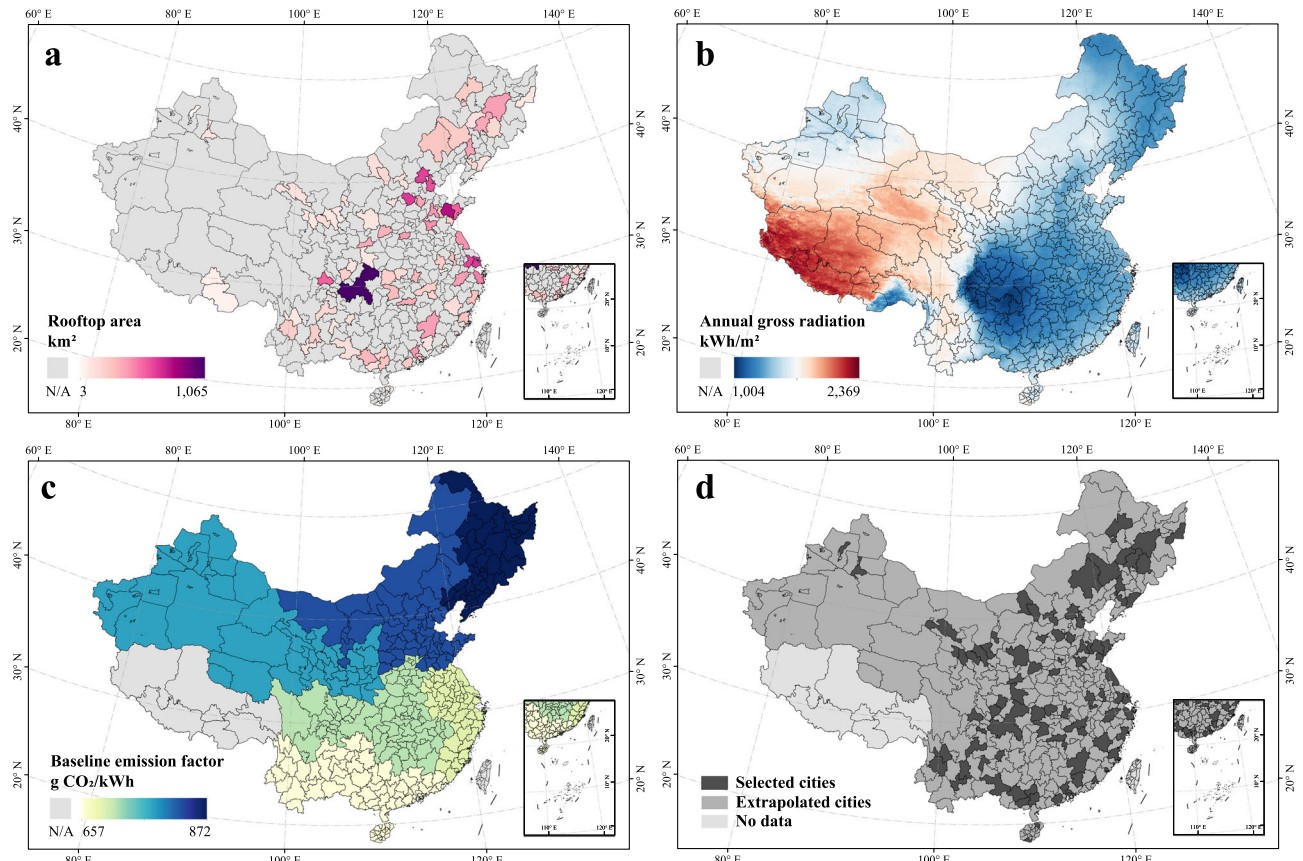

**Fig. 1 | Overview of the study area and city representations. a** Geographical scope of the vectorized rooftop data, **b** solar radiation data, and **c** grid emission data. **d** Scope of the study area extrapolated from 86 cities with measured rooftop area values to other 268 cities in China. Data Credits: All the city administrative boundaries are from Amap.

Supplementary Fig. 1). According to the comparison of the ground truth and extrapolated data, the extrapolation model we established exhibited good generalization capability and can provide refined data for the development of large-scale RPV assessment. For each 10 km² cell, the MAE was only 0.06 km², and the majority of cells had an error within −0.05–0.05 km². At the city level, a cumulative error of −26 km² was recorded in the validation area containing a total rooftop area of 2641 km², indicating that the relative error was only −1%. The relative error for each city did not exceed ±20% (Supplementary Table 9 and Supplementary Fig. 2).

### Assessment of current RPV carbon mitigation potential

To clarify the differences in location conditions that determine the RPV carbon mitigation potential, we classified the 354 cities into four groups, based on three representative indicators: rooftop area, solar radiation, and grid emissions using K-means++ clustering. This enabled us to analyze the heterogeneity of RPV potential across all the cities and to understand its causes. One-way analysis of variance (ANOVA)[39] was used to conduct pairwise comparisons between the different clusters (Supplementary Table 10). Further, 94% of these comparisons passed the significance test at the 0.01 level, indicating that the clustering results reasonably reflect the differences between the cities.

As shown in Fig. 4, we observed a clear geographic agglomeration of cities within each cluster, which indicated that the neighboring city groups tended to have similar location conditions for the application of RPVs. Cluster 1, which was geographically dispersed, contained some of the largest cities in China in terms of population size, including municipalities such as Shanghai, Beijing, and Tianjin, along with provincial capitals or regional economic centers such as Guangzhou, Suzhou, and Hangzhou. A large population size generally

implies a large building stock, and the average rooftop area of cities within Cluster 1 amounted to 498 km², approximately three times the estimated average. The cities in Cluster 2 and Cluster 3 were mainly distributed in the densely populated central and eastern regions, occupying 62% of the total rooftop area. The cities in Cluster 2 displayed a relatively clean electricity mix, with a lower average grid emission (693 g $CO_2$/kWh) than that observed in Cluster 3 (837 g $CO_2$/kWh). The cities in Cluster 4 were mainly located in the sparsely populated and underdeveloped western regions of China; among the four clusters, the average rooftop area of the cities in this cluster was the smallest (80 km²), but the solar radiation intensity was the highest (1667 kWh/m²).

The assessed installed capacity, power generation, and carbon mitigation potential of the RPVs are shown in Fig. 5. The 354 Chinese cities exhibited a total RPV reduction potential of 4 billion tons (BT) in 2020, which is nearly 70% of the carbon emissions from the electricity and heat sector. The potential of the cities ranges from 0.04 to 52 million tons (MT), with an average value of 11 MT. Thus, this study contributes to a deeper understanding of this large variation at the city level. The cities in Clusters 1–3 in the southeastern regions contributed 89% of the RPV carbon mitigation potential, which can be attributed to the large population size and abundant building stock in these areas. In particular, the top cities, in terms of carbon mitigation potentials all appear in Cluster 1, with an average potential of 29 MT. Comparing the RPV carbon mitigation potential in these cities with that of the Three Gorges Dam in 2020, the latter generated 112 TWh of electricity, directly reducing 94 MT of $CO_2$[40]. This indicates that some cities in Cluster 1, including Weifang (52 MT), Chongqing (47 MT), and Linyi (46 MT), have an RPV carbon mitigation potential roughly half that of the world's largest hydroelectric plants. Due to higher grid emissions and

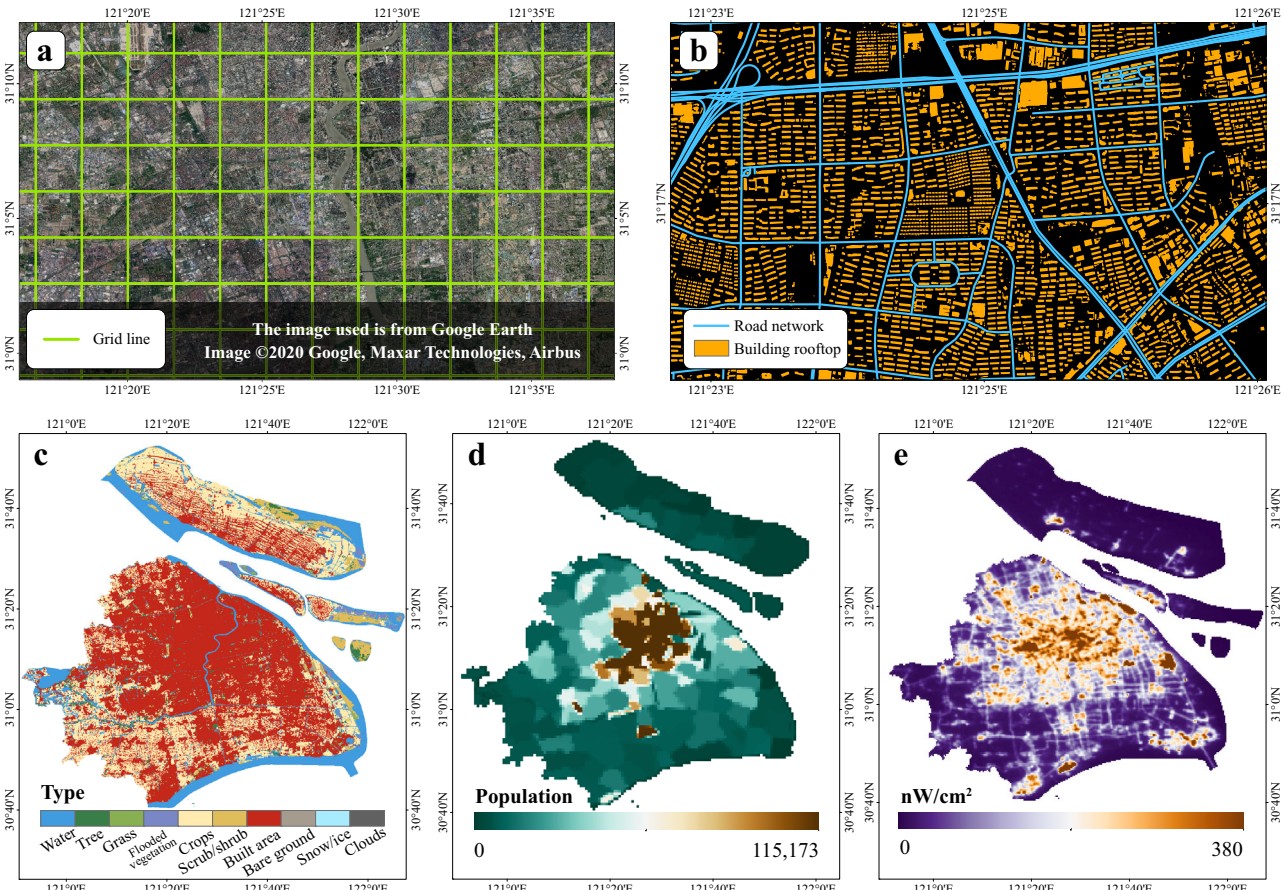

**Fig. 2 | Visual representation of the data involved in the regression analysis.**
**a** Green lines divide the basic cells of the extrapolation. **b** Orange polygons
represent the vectorized rooftop area, and blue lines represent the major road
networks. Considering Shanghai as an example, **c** land cover with the built area
colored in red indicates the potential building distribution; **d** population size with

high values in brown; and **e** night light intensity with values colors in brown
demonstrates the potential building concentration. Data Credits: NNU-SCENS Lab
Rooftop Area, OpenStreetMap, Esri Landcover, WorldPop Population, and EOG
Nighttime Light Map. All the city administrative boundaries are from Amap and the
satellite image is from Google Earth.

better solar radiation conditions, cities in Cluster 3 showed greater
average potential (11 MT) than those in Cluster 2 (9 MT). Cities in
Cluster 4, which accounted for 40% of the total study area and gen-
erally displayed optimal solar insolation endowment, had the lowest
average potential (6 MT), due to the lack of rooftop area. The RPV
carbon mitigation potential per capita population and GDP were also
estimated, showing an increasing trend from the southeast to the
northwest (Supplementary Figs. 3, 4).

**Assessment of future RPV carbon mitigation potential**
Understanding the future changes in China's RPV carbon mitigation
potential is important for a deeper understanding of its contribution to
the country's dual carbon goals. Although the available data did not
allow us to fully consider all the factors that may cause future changes,
we selected two key factors: urban land expansion and energy mix
transformation, on which we analyzed the future changes by relating
our findings to the literature. For this assessment, we chose 2030, the
year when China plans to reach its carbon peak.

China's rapid urbanization, population growth, and its popula-
tion's increasing disposable income will lead to the continuous
expansion of its building stock[41]. The increase in building stock will
further enhance the potential for RPV development, the impact of
which was not fully considered yet. Notably, urban building design and
planning in China follows uniform national standards, and urban
building density is strictly controlled by the government[42,43]. More-
over, in most existing studies, the ratio of building rooftop area to
built-up area ranged from 0.15–0.30[44]. Therefore, in this study, we

assumed that this ratio would remain stable until 2030, and thus, the
growth of rooftop areas was projected, based on the expansion of
urban land. The future urban land data was obtained from a global
study that considered different future social development pathways,
indicating that the urban land expansion rate in China will range from
9% to 14% from 2020 to 2030[45] (Supplementary Table 11). Based on this
range, this study sets low and high-speed scenarios for urban land
expansion from 2020 to 2030 (Table 1).

The potential of RPVs to reduce carbon emissions in China is
largely influenced by the country's current electricity mix. The next
question in the future is: How will the impact of RPV on carbon emis-
sions change as China's power sector goes green? The Chinese gov-
ernment has set ambitious greenhouse gas mitigation targets for the
next few decades and is committed to restructuring its energy mix.
According to an energy sector roadmap to carbon neutrality in China[46],
the current carbon intensity of China's electricity sector decreases at
an average annual rate of 1%. And with the goal of carbon neutrality, the
rate will rise to 3% in the 2020 s. Based on this projection, we set the
announced pledges (APS) scenario and stated policies (STEPS) sce-
nario for energy mix transformation from 2020 to 2030.

Depending on the urban land expansion scenarios, compared to
the 2020 data, China's rooftop area is estimated to grow by
5937–9235 km² in 2030, indicating an increase in the potential installed
capacity (416–646 GW) over the decade. In the future, RPV will
undoubtedly make a substantial contribution to China's projected
target of 400 GW of installed PV capacity by 2030. In the APS energy
mix transformation scenario, where the grid emissions drop sharply,

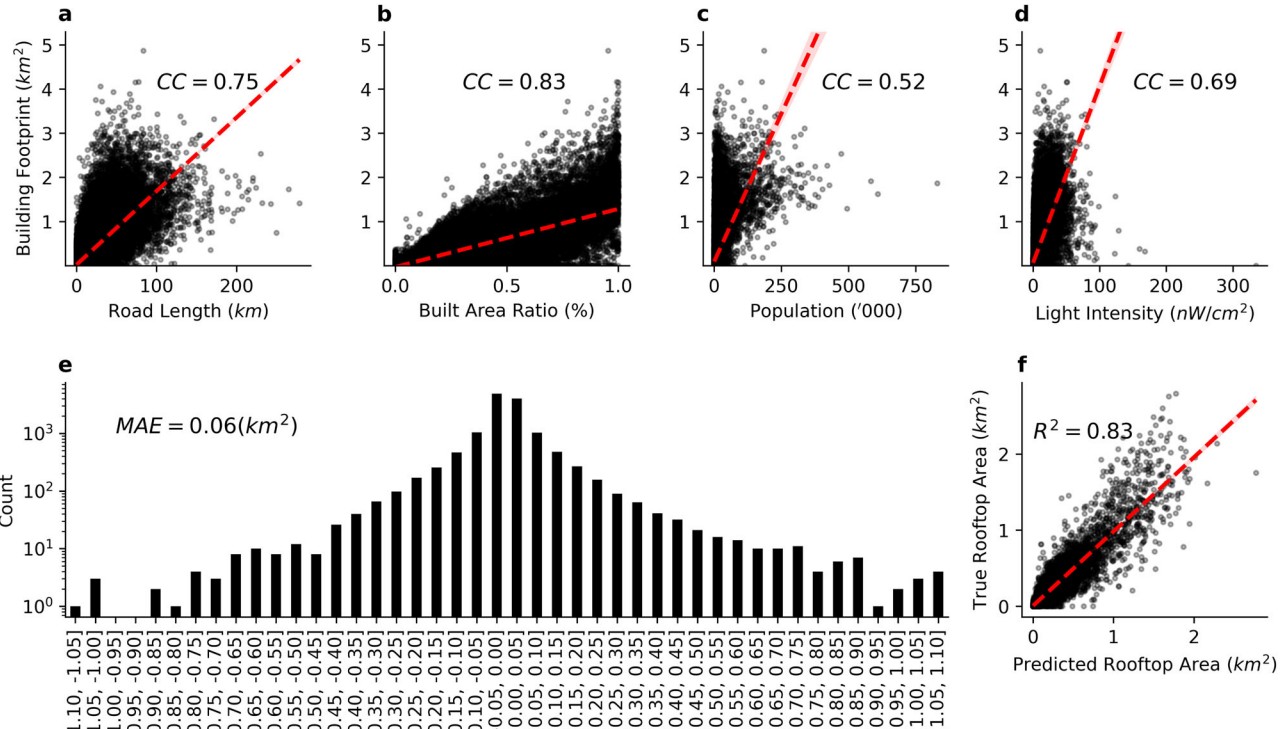

**Fig. 3 | Relationship between inputs, distribution of errors, and performance of the extrapolation model. a** Road length, **b** proportion of built-up area, **c** population size, and **d** nighttime light intensity relative to the rooftop area. The Pearson correlation coefficients (CC) for these distributions are 0.75, 0.83, 0.52, and 0.69, respectively (significant at the 0.0001 level). **e** When the trained model was used to predict the validation dataset, the histogram distribution of prediction errors was bell-shaped, with most of the grids lying within −0.05–0.05 km², and with a mean absolute error (MAE) of 0.06 km². **f** The machine learning regression model based on the random forest approach provided sufficient prediction accuracy with an $R^2$ value of 0.83.

the carbon mitigation potential of the RPV is reduced by more than 20%, even when considering the potential increase in the installed capacity. However, in the STEPS scenario, the grid emissions remained at a high level, with the final estimated potential being slightly higher or lower than that in 2020. This also means that for China, the RPV mitigation potential will gradually reduce with the penetration of renewable energy and the decarbonization of the power sector. However, RPV may still be considerable over the next decade.

### Exploitation of RPV carbon mitigation potential

To understand how to exploitation the theoretical potential for the transformation of China's energy mix and to help achieve the country's carbon mitigation targets, we conducted in-depth quantitative and qualitative analyses (Fig. 6). The quantitative analysis ranked the priority of cities to develop RPVs in quartiles, using the volume and intensity of carbon mitigation as indicators. The qualitative analysis explored the development of the theoretical potential, in terms of the current PV installation, energy consumption, and carbon emissions. The analyses were conducted on a provincial basis, deepening the understanding of the transition from theory to practice.

The volume of carbon mitigation follows a top-down perspective, where governments are typically more concerned with whether the total local potential can support the goal of achieving the established macro targets (Fig. 6a). The intensity of carbon mitigation follows a bottom-up perspective, where businesses and individuals are typically more concerned with whether the potential per unit can provide high benefits (Fig. 6b). Taking, for example, the 11 prefectures in Shanxi Province, Yuncheng contributed the most RPV carbon mitigation potential, accounting for 18% of the provincial total, which is more capable of relieving the provincial carbon mitigation pressure. Meanwhile, Shuozhou contributed the highest RPV mitigation intensity, with 4% higher than the provincial average, which is more capable of

achieving higher environmental benefits when implementing an RPV project. In addition, it is crucial to comprehensively consider the volume and intensity of carbon mitigation. Cities with both low volume and intensity of carbon mitigation potential (such as Tongren, Zigong and Anshun) may face more pressure to implement RPV (Fig. 6c).

To demonstrate the gap between the theoretical and existing installed capacity, the cumulative DPV installed capacity data for the whole country in 2018 was used[47], along with the 80% share of the RPVs in the DPVs[11]. The comparison shows that 73% of the provinces/municipalities have developed <1% of their installed potential (Fig. 6d). This indicates that, although China has become a leading country in absolute global PV production and installed capacity, there is still plenty of room for further market expansion. To demonstrate the comparison between the theoretical power generation and current electricity consumption, provincial electricity consumption data for 2020 were used[48]. The results indicated that 80% of the provinces/municipalities had power generation potential that exceeded half of the local electricity consumption (Fig. 6e). However, the development of RPVs in some regions may not be enough to supplement the local electricity demand. To demonstrate the comparison between theoretical carbon mitigation and the current carbon emissions, local carbon emissions from the electricity and heat sector in 2019 were used[49]. The results indicated that, in the northern and northeastern power grids, the mitigation of carbon emissions per unit of RPV power generation will be more pronounced (Fig. 6f). From the perspective of power decarbonization, in provinces/municipalities with high coal dependencies such as Ningxia and Shanxi, where manufacturing and heavy industry are dense, the development of DPVs can better facilitate local power transformation.

## Discussion

This study provides a national assessment of RPV carbon mitigation potential at the city level in China. Based on multisource

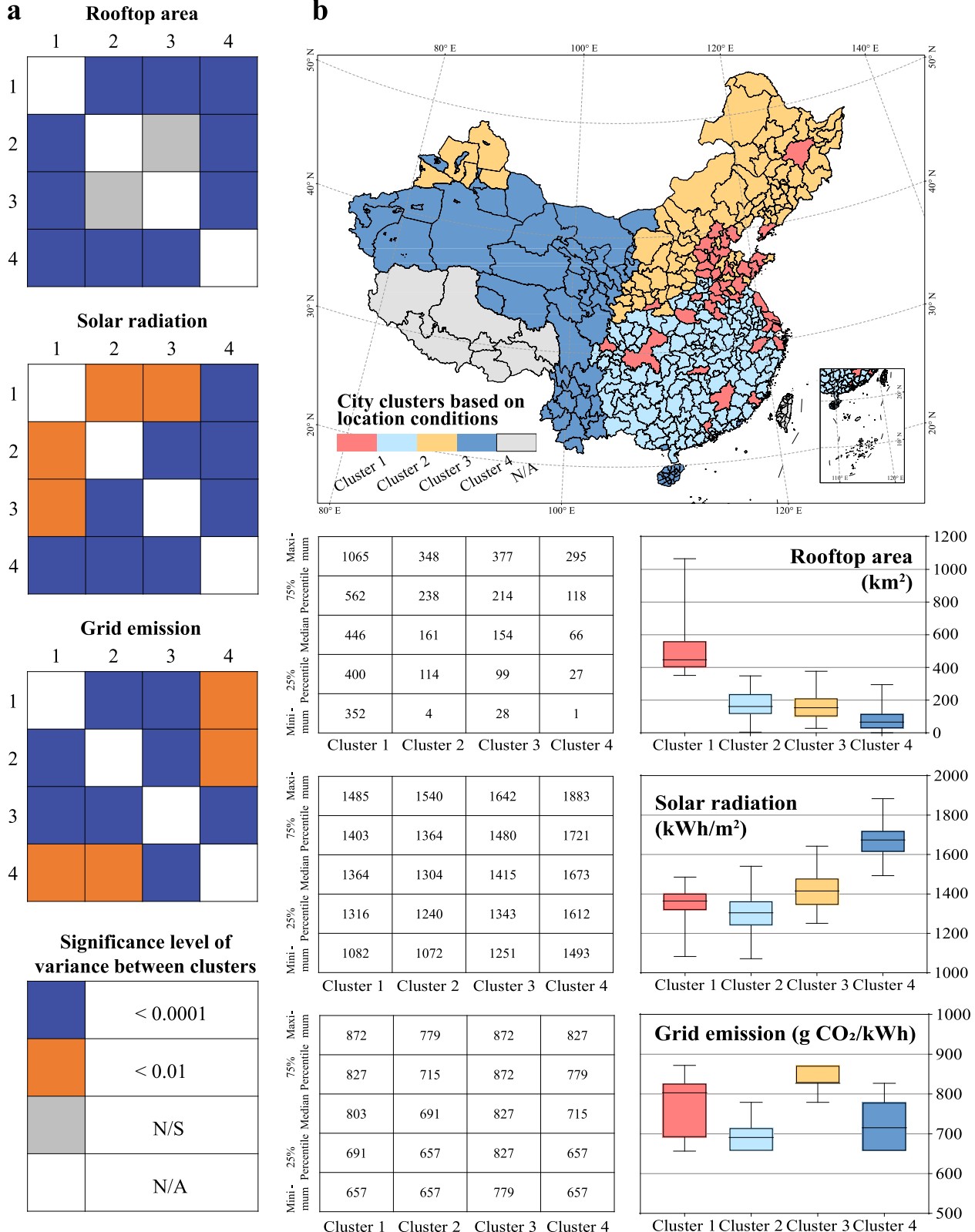

**Fig. 4 | Clustering analysis of cities, based on location conditions that determine the RPV carbon mitigation potential. a** K-means++ clustering was performed for the three types of location conditions that determine the RPV carbon mitigation potential, and the 354 cities were categorized into four clusters with statistically significant differences in rooftop area, solar radiation, and grid emission. **b** The map and box plot display the spatial distribution and location condition characteristics of cities in each cluster, respectively. The east–west divisions portray the differences in the rooftop areas and solar radiation intensity, and the north–south divisions portray the differences in the grid-level emissions. Data Credits: All the city administrative boundaries are from Amap.

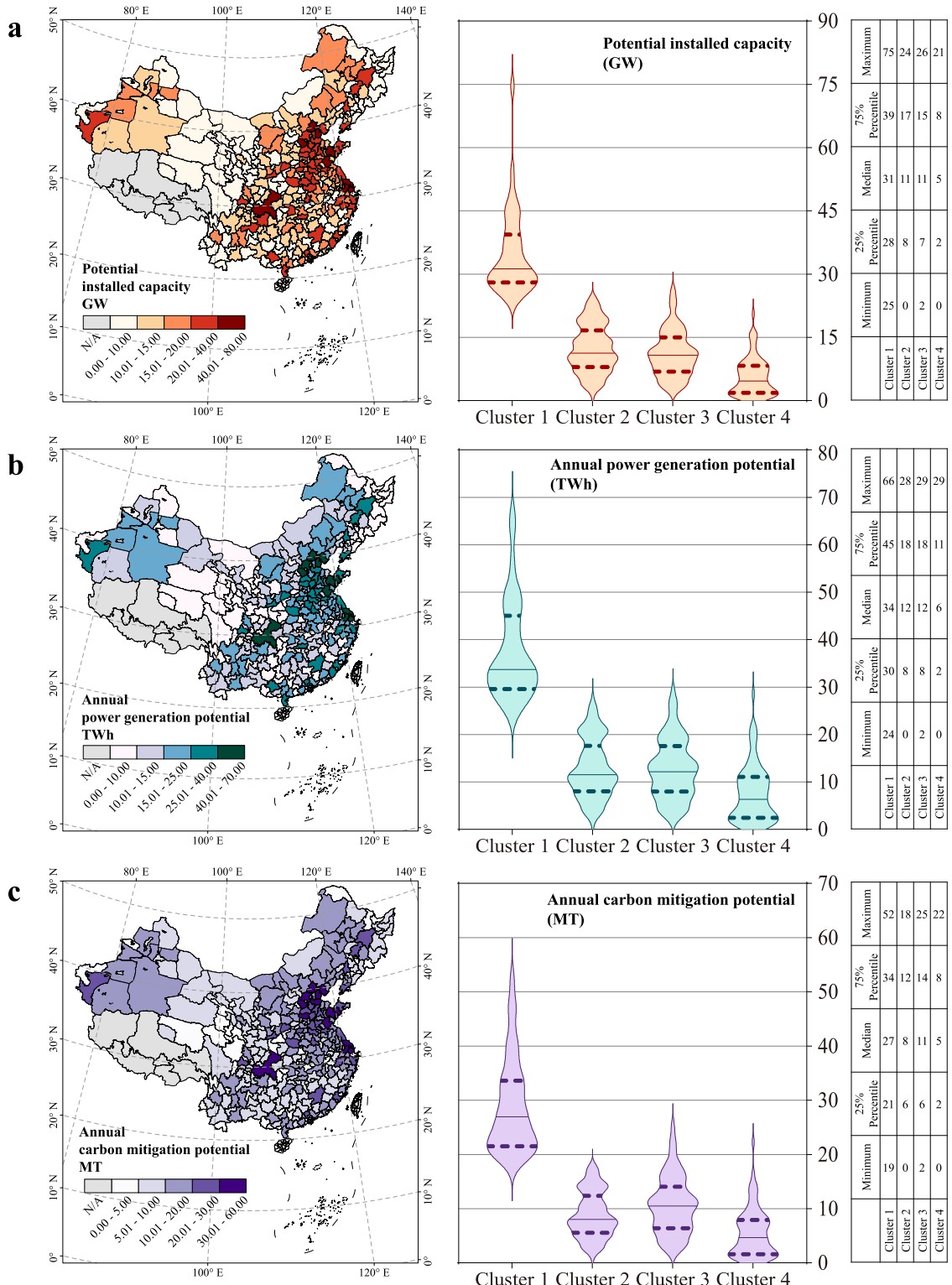

**Fig. 5 | Key indexes of the RPV carbon mitigation potential assessment for 354 Chinese cities in 2020. a** Geographical distribution of the RPV installed potential, **b** power generation potential, and **c** carbon mitigation potential for the 354 cities. The left and right sides of the figure represent the results aggregated at the city and cluster levels, respectively. The cities in Cluster 1 generally have the largest population and a productive economy, representing the largest installed and power generation scale, with a contribution of 29% of the total carbon mitigation potential, while Cluster 4 accounted for only 11%. Data Credits: All the city administrative boundaries are from Amap.

**Table 1 | Future changes in rooftop carbon mitigation potential under different urban land expansion and power mix transformation scenarios**

| Power mix transformation scenarios | Emission factor decline rate from 2020 to 2030 | Urban land expansion scenarios | Rooftop area growth rate from 2020 to 2030 | Carbon mitigation potential in 2030 (MT $CO_2$) | Carbon mitigation potential change from 2020 to 2030 (MT $CO_2$) |
|---|---|---|---|---|---|
| STEPS | 10% | Low speed | 9% | 3730 | −72 |
| | | High speed | 14% | 3901 | 99 |
| APC | 30% | Low speed | 9% | 2901 | −901 |
| | | High speed | 14% | 3034 | −768 |

heterogeneous geospatial data and machine learning regression method, the key data and computational limitations generally faced in existing estimation are addressed. As part of the assessment, we created (1) a national rooftop area dataset with high-resolution of 10 km² grid, (2) a national RPV carbon mitigation potential dataset at the city level. Furthermore, the scenarios of urban land expansion and power mix transformation were considered, in order to understand the variation in RPV carbon mitigation potential by 2030. The current status of installed capacity, energy consumption, and grid emissions in China was introduced to understand how to optimally exploit the potential. Despite various favorable policies aimed at promoting DPV development, local adoption of RPVs is still quite limited relative to long-term national goals, and more targeted measures are required to increase public awareness from a regional perspective. Thus, the findings of this study can help identify key factors for local RPV development through trend simulation, impact analysis, and regional comparisons.

Our assessment has important implications for addressing the dual challenges of sustainable development and climate change. First, the total RPV carbon mitigation potential of the 354 cities in 2020 was determined (4 BT), which amounted to nearly 70% of the carbon emissions from the electricity and heat sector. This highlighted an important aspect of solar resource development, suggesting a greater use of building rooftops for the development of DPV systems in the context of dual carbon goals; this can help China because it has limited land space available for PV installation. Second, the regional analysis indicated that approximately 89% of the potential was located in the densely populated southeastern regions. This suggested that the eastern cities, having higher demographic dividends, could be the first to decarbonize the power generation infrastructure by deploying RPVs. Third, as for the sparsely populated northwestern cities, they are considered more suitable for centralized PV development. Fourth, for highly industrialized cities with high coal dependency, it is more important and necessary to develop RPVs and accelerate the construction of a new energy-based power system.

Our city-level assessment provides quantitative indicators of the volume and intensity of carbon mitigation from different perspectives, both top-down and bottom-up. This can help local governments, companies, and individuals to identify locations suitable for the rapid deployment of RPV systems and to promote energy justice. The assessment also provides insightful findings and detailed datasets of RPV potential, which can improve future models of carbon-neutral scenarios and underpin important national energy policies. This will undoubtedly benefit the exploration of the possibilities of a sustainable and inclusive low-carbon future.

The estimation results in this study portrayed the theoretical maximum carbon mitigation potential of RPV systems in China. In our main analysis, the results were based on the assumption that 35% of the rooftops were available for RPV installation. In the current literature, reducing the total rooftop area to the available rooftop area is typically accomplished by a rooftop scaling factor that represents the loss of rooftop area due to orientation, slope, and obstacles, etc. Although the conversion factor currently used has covered the above aspects, it is a quantification of the average situation at the national level. City-level

availability may vary, for which we conducted supplementary studies to provide more understanding (Supplementary Note 7 and Supplementary Tables 15–17). In addition, we provided the variations in the RPV carbon mitigation potential as an uncertainty analysis for the combination of the rooftop availability and panel efficiency.

Notably, in conventional power systems, which are dominated by fossil fuel-based generation, the power output can be adjusted according to hourly, weekly, and seasonal fluctuations in load. However solar energy availability is dependent on weather conditions, making this type of energy inflexible. To increase the future penetration of RPVs, there is a need in future studies to further explore how to improve the flexibility of RPV energy production. For example, combining RPVs with energy storage technology to achieve continuous power supply at night and reduce stress on the power grid. Accordingly, the complementarity between renewable energy sources is another possible solution to enhance the stability of the power grid. In fact, the Chinese government is making continuous efforts to advance the efficient future deployment of PV systems. Most Chinese provinces are currently promoting policies to equip PV energy storage facilities at no less than 10% (and in some cities even 20%) of PV installed capacity[50,51]. Additionally, more and more renewable energy pilots are being built to explore new modes of solar-wind-hydro integration.

Economic factors are another important aspect to exploit the technical potential of RPV. Although the scope of this work does not include a detailed assessment of the economic feasibility at city scale, the existing literature has provided a basic understanding. In 2020, the initial investment cost of commercial and industrial DPV systems in China is 3.38 Yuan/W. The levelized cost of energy (LCOE) for DPV systems under the full investment model is 0.17, 0.20, 0.26, and 0.31 Yuan/kWh at 1800, 1500, 1200, and 1000 equivalent utilization hours, respectively[52]. Without subsidies, commercial and industrial DPV in all 344 Chinese cities has achieved 100% user-side grid parity[10], and household DPV in 86% cities has been shown to be economically viable[11]. In addition, China has implemented a series of large-scale initiatives to systematically deploy PV projects to alleviate poverty in rural areas[53].

Even with its limitations and shortcomings, the current study provides scholars with a national-level dataset to support future work. Future studies could improve the accuracy of rooftop area prediction by introducing more comprehensive explanatory variables and higher resolution training data, thus enabling a more detailed national-level assessment of RPV carbon mitigation potential. In order to better understand the environmental benefits that RPV systems can provide in the future, detailed assessments of the costs associated with fulfilling their abatement potential are also relevant. Particularly, the extensive integration of PV systems with urban infrastructure is crucial to diversify the exploitation of solar resources, and more forms of PV systems can be considered in the next studies.

## Methods
### Assumptions and system parameters
We made unified assumptions for the RPV systems involved in this study. According to recent literature review[54], the system efficiency

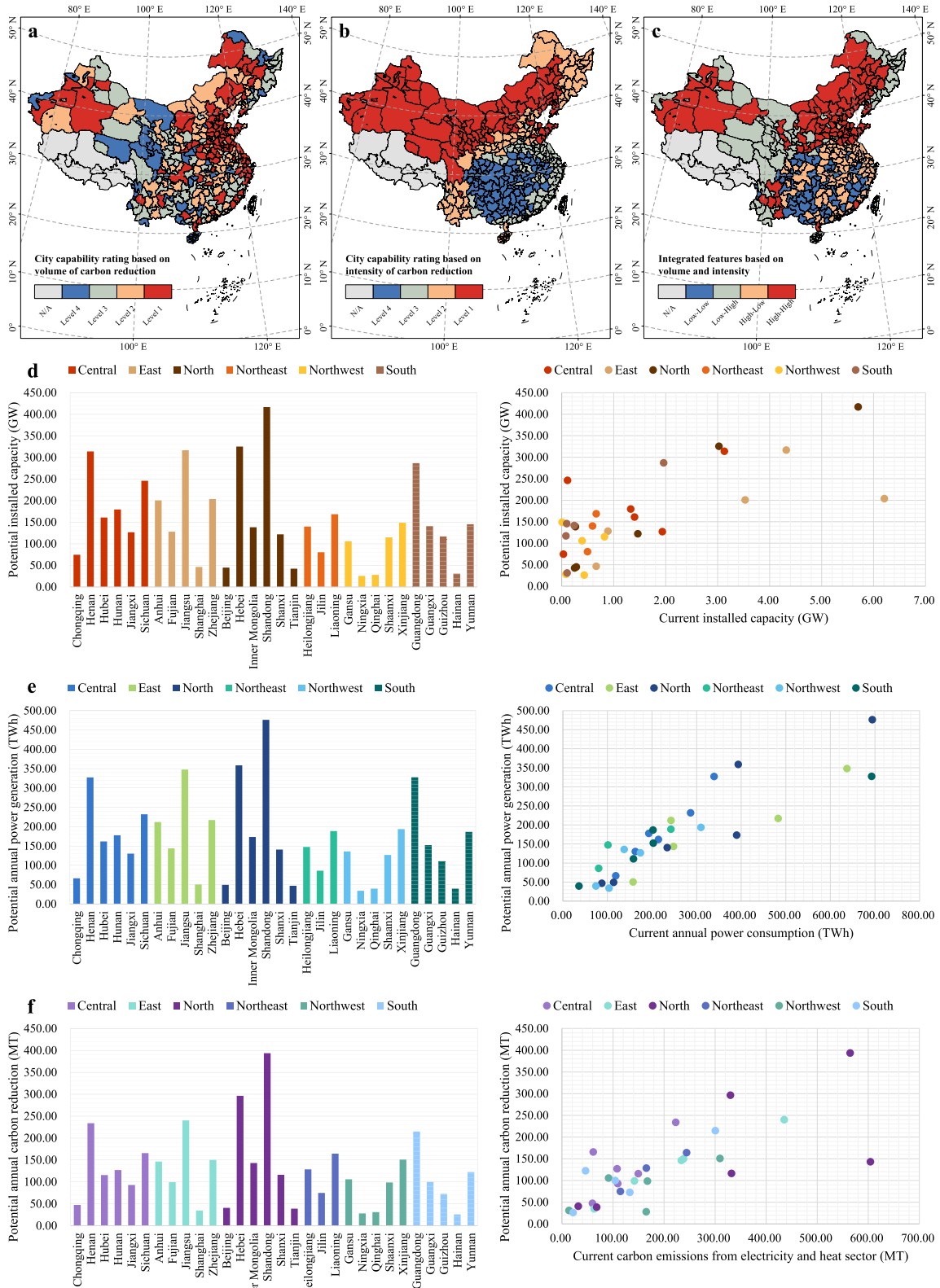

**Fig. 6 | Regional analysis for the sustainable development of RPVs. a** City capability rating based on the volume of RPV carbon mitigation. **b** City capability rating based on the intensity of RPV carbon mitigation. Levels 1 to 4 represent decreasing capability. **c** Integrated features of carbon mitigation volume and intensity, with the High-High pattern representing the city in the top 50% for both indicators. **d** Comparison of theoretical installed capacity and existing cumulative installed capacity. **e** Comparison of theoretical power generation and current electricity consumption. **f** Comparison of theoretical carbon mitigation and current carbon emissions. Data Credits: All the city administrative boundaries are from Amap.

correction parameters of module efficiency, light utilization, component surface contamination, conversion efficiency, inverter efficiency, line loss, transmission efficiency, and PV system availability were determined (Supplementary Table 12). The average energy efficiency ratio of the PV system was calculated as 0.8. According to the current technical level of the PV industry[55] the scale and performance parameters of PV applications were determined (Supplementary Table 13), including a PV conversion efficiency of 20%, and a rated power per unit area of the PV panel of 200 W. We assumed that all the PV panels in a system were fixed in a horizontal position.

## Regression analysis of rooftop area

To relate the existing rooftop area of 86 cities to the whole country, a regression model was developed. Road length, built-up area, population size, and night light intensity were selected as the explanatory variables. These variables were counted in grid cells and, finally, aggregated into structured tables. The sample data of the 86 cities were first normalized and divided, with 90% of the data used as the training data and 10% used as the test data. For the regression method, we selected the random forest algorithm[56]. In general, the random forest algorithm creates one model by combining the results obtained from several well-built models, and the final prediction is determined by averaging the prediction results of all the models. The basic steps followed in this study are explained below:

(1) Sampling: From the training dataset $S$, $K$ datasets were generated through sampling, with replacement, and each dataset was trained to generate a decision tree.
(2) Growing: At each sub node, $m$ features were randomly selected from $M$ attributes for growth based on Gini metrics.
(3) The test dataset was able to validate the model effect and generalization ability, as it was not involved in modeling. A 10-fold cross-validation was performed with the test dataset, and the hyperparameters of the algorithm, such as the number of decision trees, were determined.
(4) Prediction: For the new dataset, the average of the prediction results of all the decision trees was considered as the final output.
(5) The prediction of the new dataset was performed using the trained model, and the average of the prediction results of all the decision trees was considered as the final output.

## Assessment of RPV carbon mitigation potential

The potential installed capacity of RPV systems is critical for the estimation of their power production and the corresponding carbon mitigations. For the installed capacity involved in this study, it was assumed that 35% of the building rooftops were available for the deployment of RPV systems[57]. Specifically, the conversion factor of rooftop availability has considered the potential impacts of building social function, geometric typology, slope & orientation, structural quality, economic cost, and shadow & obstacle.

The potential installed capacity, $P_{\text{install}}$, was calculated using Eq. (1), as follows:

$$P_{\text{install}} = P_R \times S \times C_{\text{RA}} \tag{1}$$

where $P_R$ is the rated power of the PV panel per unit area, $S$ is the rooftop area, and $C_{\text{RA}}$ is the conversion factor for calculating the available rooftop area for PV installation.

The annual power generation, $P_{\text{power}}$, of the RPV system was estimated using Eq. (2):

$$P_{\text{power}} = P_{\text{solar}} \times C_{\text{PV}} \times K \tag{2}$$

where $C_{\text{PV}}$ is the conversion efficiency of the PV panel, $K$ is the overall efficiency of the PV system, and $P_{\text{solar}}$ is the rooftop solar potential,

which can be further obtained by Eq. (3):

$$P_{\text{solar}} = \sum_{i=1}^{n_1} (S_i \times GI_i) \tag{3}$$

where $S_i$ is the area of the $i$th rooftop, $GI_i$ is the annual surface solar radiation received by the $i^{\text{th}}$ rooftop, and $n_1$ is the total number of rooftops.

In this study, we focused on the power generation stage of the power system without considering any other life cycle stages (such as facility construction, transportation, and recycling). This is because the life-cycle carbon emissions of RPV systems are much smaller than the emissions prevented during the operational stage[54]. The baseline emission factor of China's power grids, which is used to account for carbon mitigation from renewable energy projects, consists of the operating margin (OM) and the build margin (BM) factors. The OM and BM factors measure the carbon emission intensity of existing power plants in the grid and newly-built power plants, respectively. The combined marginal (CM) factor of the weighted average of the OM factor and the BM factor is generally used to express the carbon emissions intensity of the grid during electricity production. Since a PV system is generally considered clean during the electricity production, the CM factor, $EF_{\text{grid1,CM},y}$, can be used as the carbon mitigation factor, $ERF_{\text{PV}}$, of the RPV system, to measure the carbon emissions prevented by power generation using RPVs. The calculation was performed according to Eq. (4)[58], as follows:

$$ERF_{\text{PV}} = EF_{\text{grid,CM},y} = EF_{\text{grid,OM},y} \times W_{\text{OM}} + EF_{\text{grid,BM},y} \times W_{\text{BM}} \tag{4}$$

where $EF_{\text{grid,OM},y}$ is the OM factor, and $EF_{\text{grid,BM},y}$ is the BM factor. $W_{\text{OM}}$ is the weight of the OM factor, and $W_{\text{BM}}$ is the weight of the BM factor; the values were set to 0.75 and 0.25, respectively[56].

According to the carbon mitigation factors of the RPV system for different power grids, carbon mitigation, $P_{\text{mitigation}}$, was calculated according to Eq. (5):

$$P_{\text{mitigation}} = P_{\text{power}} \times ERF_{\text{PV}} \tag{5}$$

where $P_{\text{power}}$ is the annual power generation of the RPV system.

## Clustering analysis for location conditions

For the three location conditions: rooftop area, solar radiation, and grid emissions, which affect the RPV carbon mitigation potential, the K-means++ clustering algorithm was used to classify the 354 cities into different clusters. The steps of the K-means++ algorithm are as follows[59]:

(1) Randomly select a center $u_1$ between data points.
(2) For each data point $x$ that has not yet been selected, calculate $\sum_{i=1}^{i=j} d(x, u_i)$, the distance between $x$ and the closest center that has been selected.
(3) A new data point is randomly selected as the new center using a weighted probability distribution, where the probability of the selected point $x$ is proportional to $\sum_{i=1}^{i=j} d(x, u_i)$.
(4) Repeat steps 2 and 3 until $k$ centers have been selected ($j = k$).
(5) Calculate the distance between the sample and each initial center, and assign the sample to the corresponding cluster according to the distance.
(6) Calculate the new center, i.e., the mean of each cluster $1/|C_i| \sum_{x \in C_i} x$.
(7) Repeat steps 2 and 3 until all the clustering centers are stable.

# Data availability

The data that support the findings of this study are provided in Supplementary Data 1. The vectorized rooftop area data for 90 cities in China[60] are available at https://doi.org/10.11888/Geogra.tpdc.271702.

Other data sources that are free to use are provided in the supplementary materials (Supplementary Table 14).

## Code availability

The Python scripts corresponding to rooftop area extrapolation and urban clustering are available at https://github.com/ChanceQZ/Core-code-of-carbon-mitigation-x-rooftop-solar-pv. The code can also be accessed via https://doi.org/10.5281/zenodo.7766125 in Zenodo[61].

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

## Acknowledgements
M.C. acknowledges support from the International Research Center of Big Data for Sustainable Development Goals (CBAS2022GSP08). G.N.L. acknowledges support from the key program of the National Natural Science Foundation of China (Grant No.41930648). L.X.T. acknowledges support from the National Key R&D Program of China (Grant No. 2020YFA0608601). J.Y.Y. acknowledges support from the Swedish Knowledge Foundation (KKS)'s Flexibility project, the HKPolyU (BDCH) project FUES, and the RISUD project (P0000148) (Data-driven solutions for decarbonizing transportation sector by coupling renewable energy, energy storage and smart EV-charging).

## Author contributions
Z.X.Z. and M.C. designed the study. Z.X.Z. performed the analysis and wrote the first version of the manuscript. M.C. coordinated the work and guided the experiments. T.Z., R.Z., and F.Z. provided technical guidance and reviewed the manuscript. Z.Q., Y.Y., and K.Z. assisted with data processing. P.S., K.C.W., and Y.X.P. assisted with quality control and reviewed the manuscript. L.X.T., G.N.L., and J.Y.Y. condensed experimental ideas and reviewed the manuscript. All the authors contributed to the final version of the manuscript.

## Competing interests
The authors declare no competing interests.
