## [Peer Review File · Nature Communications]

Carbon mitigation potential afforded by rooftop photovoltaic in ChinaREVIEWER COMMENTS

Reviewer #1 (Remarks to the Author):

The paper is interesting and well written and the most interesting result is in the methodology to evaluate the potential of roof mounted PV systems at state level in China to estimate the carbon mitigation potential.

The paper deserves to be published but there are some issues that should be clarified before the paper can be accepted.

1) The estimation of the rooftop surface has been made by analysing the maps with the outline of buildings, streets and squares. Nonetheless, it is not clear if the type of building has been identified (residential, commercial, industrial). This would be very important for a better calculation of the available surface. It might happen that most of high rise buildings cannot be used since HVAC plants are located on top, as well as many other service equipment reducing significantly the free surface for the installation of the PV systems. This would make the methodology much more precise in calculating the available surface.

2) another important point is what direction is the roof facing. If all the surface have been calculated, there could be a significant overestimation of the potential for installation.

3) Have squares and open spaces such as parking spaces, been considered for potential use ?

4) The overall installed PV peak power is huge but it could not be possible just to consider the area and the yearly generated energy to make the possible installation sustainable in terms of balancing power generated during the sunlight hours. In case of such a significant amount of PV power, is it possible to estimate the energy storage needs ?

5) A rough estimate of the overall investment costs and of the energy and materials inventory for the production of such a huge amount of PV systems would be helpful to understand the cost of CO2 mitigation with a similar deployment of solar systems at worldwide level.

Reviewer #2 (Remarks to the Author):

This is a very well prepared and presented paper, dealing with an important area addressing the carbon mitigation problem for the world, but in particular for China. China, as the authors note, is the leading country in total emissions (though the U.S. is certainly ahead on a per capita basis). The authors have presented a very good argument and rationale for their study and analysis. This reviewer is very impressed with this manuscript, and the reviewer is strong in the recommendation that the paper be published. The review has only a few minor comments, questions or recommendations.

1. The paper analysis and approach is excellent. The authors have done a very commendable job in providing the explanations and the results. There do not seem to be any weak points in the analysis.

2. Is there an estimate of what portion of the rooftops are currently suitable for solar (PV) utilization in China? And what impact this would have to energy and carbon issues. (The carbon part is covered, but I was unsure about the percentage of solar-suitable roofs and orientations.)

3. How does the roof orientation etc. (solar suitability) compare with other countries. This might provide an argument for a more world-effort with rooftop installations. Could a table be added to partially summarize this?

3. Is the built environment in general in China durable for this massive effort? The reason for asking is that in many countries, the most impact would be in the poorer areas (e.g., slums or favelas) where the support and security for PV (for example) may not be sufficient. Are there comments on this?

4. Are the cost comparisons between large plants and solar rooftops compelling?

This is just a very impressive paper and should be published. The authors' current version is suitable for publication and does not have to be changed for the comments above (unless the authors would

like to include further ideas.) The paper is technically strong--and covers a topic that should be addressed.

Dear Reviewers,

Thank you very much for your time involved in reviewing the manuscript and your very encouraging comments on the merits.

Reviewer #1:

The paper is interesting and well written and the most interesting result is in the methodology to evaluate the potential of roof mounted PV systems at state level in China to estimate the carbon mitigation potential.

The paper deserves to be published but there are some issues that should be clarified before the paper can be accepted.

Author:

The authors appreciate Reviewers 1 for the clear and detailed feedback and hope that the explanation has fully addressed all of your concerns. We will then discuss each of your comments in turn and our corresponding responses.

Comment 1:

The estimation of the rooftop surface has been made by analysing the maps with the outline of buildings, streets and squares. Nonetheless, it is not clear if the type of building has been identified (residential, commercial, industrial). This would be very important for a better calculation of the available surface. It might happen that most of high-rise buildings cannot be used since HVAC plants are located on top, as well as many other service equipment reducing significantly the free surface for the installation of the PV systems. This would make the methodology much more precise in calculating the available surface.

Response 1:

The conversion from rooftop area to available rooftop area in this article is done by using a conversion factor. According to a study of renewable energy availability in 139

countries around the world, the conversion factor for rooftop area in China was set at 35%¹. The reference above has provided a comprehensive analysis of rooftop availability at the national level. Specifically, the conversion factor currently used has considered building social function, geometric typology, slope & orientation, structural quality, economic cost, and shadow & obstacle.

For social function, residential, industrial, and commercial/government/institutional buildings were classified. For geometric typology, flat and pitched rooftops were classified. For slope & orientation, pitched rooftops are less available than flat rooftops; and steep rooftops are less available than gently rooftops, considering the efficiency of sunlight reception. For structural quality, the installed capacity limitations due to different building masses were considered. For economic cost, differences in deployment capabilities due to affordability were considered. For shadow & obstacle, shadows caused by mutual shading between buildings of different heights; and barriers caused by fire escapes, etc. were considered.

Although the conversion factor currently used has been integrated for numerous aspects, it is a quantification of the average situation at the national level. City-level availability may vary, but due to the limitations of current data sources (e.g., policy-related data and three-dimensional data are difficult to obtain across more than 300 Chinese cities), we are unable to further refine the currently used factors to the city level.

However, in order to provide the readers with more information, we selected four representative cities with different administrative levels and implemented an alternative approach to clarify the differences in city-level rooftop availability based on the data sources that we can obtain. In our supplementary experiments, four major aspects, including building social function, geometric typology, slope & orientation, and shadow & obstacle, are considered.

Adopting the assumptions of the average case, the results of the supplementary study showed that the conversion factor of rooftop availability at the city level ranged from 0.32 to 0.35, which is similar to the factor at the national level currently used. In

developed cities with higher administrative levels, rooftop availability is lower due to the increased proportion of unavailable land with cultural heritage and public facilities properties. In addition, even adopting the worst-case assumptions, the conversion factor is between 0.24 and 0.26.

Revision 1:

- Page 13, Line 5-13 and Page 14, Line 37-40: We have refined the description of accounting for available rooftop area, clarifying the aspects that have been considered for the currently used conversion factors.
- Supplementary Note 7: We have added the estimation process and results of the city-level rooftop availability conversion factor.

Comment 2:

Another important point is what direction is the roof facing. If all the surface have been calculated, there could be a significant overestimation of the potential for installation.

Response 2:

According to your comments, we have provided additional explanations on the accounting of available rooftop area. As mentioned in response 1, the conversion factors currently used already take into account the availability limitations due to slope and orientation at the national level. However, to clarify the specifics at the city level, we have chosen representative cities for supplemental study.

Revision 2:

- Page 13, Line 5-13 and Page 14, Line 37-40: We have refined the description of accounting for available rooftop area, clarifying the aspects that have been considered for the currently used conversion factors.
- Supplementary Note 7: We have added the estimation process and results of the city-level rooftop availability conversion factor.

Comment 3:

Have squares and open spaces such as parking spaces, been considered for potential use?

Response 3:

In China, a densely populated country with a large building stock, RSPV with no additional land occupation offers significant advantages. The Chinese government has also developed a series of supportive policies to promote the application of RSPV. RSPV is considered to have a promising future in achieving China's dual carbon goals. Therefore, this work focuses on building rooftops and aims at a quantitative assessment of their solar potential.

However, it is still a very important aspect to consider the integration of PV systems with other urban infrastructures and some unused land. We have already done some preliminary work to explore these aspects, such as using road noise barriers and highways to deploy PV systems^{2,3}. In the future, we will concentrate on more forms of PV applications and evaluate the city-level potential more extensively.

Revision 3:

- Page 13, Line 40-42 and Page 14, Line 1-2: We have added the integration of PV systems with other urban infrastructures to the future study.

Comment 4:

The overall installed PV peak power is huge but it could not be possible just to consider the area and the yearly generated energy to make the possible installation sustainable in terms of balancing power generated during the sunlight hours. In case of such a significant amount of PV power, is it possible to estimate the energy storage needs?

Response 4:

Thank you for pointing out. For intermittent energy sources like solar, it is necessary to maximize their utilization through additional energy storage. Our current assessment

focuses on the annual amount of the theoretical potential. However, in the discussion section, the support measures that may be needed to consume such a large amount of energy are presented.

In fact, to achieve power balance in real time, hourly power demand and supply curves should be analyzed. However, this is beyond the scope of this work. In addition, city-scale hour-by-hour supply and demand data are currently unavailable. To clarify the critical issue of energy storage requirements, this article provides a basic reference by taking into account the practicalities of China's rollout of RSPV projects.

Revision 4:

- Page 13, Line 18-25: We have detailed the support measures needed to consume such a large amount of energy and added China's current PV energy storage policy to the discussion.

Comment 5:

A rough estimate of the overall investment costs and of the energy and materials inventory for the production of such a huge amount of PV systems would be helpful to understand the cost of CO2 mitigation with a similar deployment of solar systems at worldwide level.

Response 5:

This work focuses on assessing the technical potential of RSPV. However, in terms of implementation, economic feasibility is a vital aspect. Although a detailed economic assessment is beyond our scope, we have combined existing studies that have addressed this issue in detail to enrich our paper.

Revision 5:

- Page 13, Line 26-35: We have added a discussion of the economic feasibility of RSPV deployment, providing references for its investment cost and levelized cost.

Reviewer #2:

This is a very well prepared and presented paper, dealing with an important area addressing the carbon mitigation problem for the world, but in particular for China. China, as the authors note, is the leading country in total emissions (though the U.S. is certainly ahead on a per capita basis). The authors have presented a very good argument and rationale for their study and analysis. This reviewer is very impressed with this manuscript, and the reviewer is strong in the recommendation that the paper be published. The review has only a few minor comments, questions or recommendations.

This is just a very impressive paper and should be published. The authors' current version is suitable for publication and does not have to be changed for the comments above (unless the authors would like to include further ideas.) The paper is technically strong--and covers a topic that should be addressed.

Author:

Thank you for recognizing our work. The authors would like to thank Reviewer 2 for the hard work in reviewing this article. We have carefully revised the manuscript according to your valuable advice and provided point-by-point responses as follows.

Comment 1:

The paper analysis and approach is excellent. The authors have done a very commendable job in providing the explanations and the results. There do not seem to be any weak points in the analysis.

Response 1:

Thank you for your high praise. We have addressed any other parts you have raised.

Comment 2:

Is there an estimate of what portion of the rooftops are currently suitable for solar (PV) utilization in China? And what impact this would have to energy and carbon issues.

(The carbon part is covered, but I was unsure about the percentage of solar-suitable roofs and orientations.).

Response 2:

The conversion from rooftop area to available rooftop area in this article is done by using a conversion factor. According to a study of renewable energy availability in 139 countries around the world, the conversion factor for rooftop area in China was set at 35%¹. The reference above has provided a comprehensive analysis of rooftop availability at the national level. Specifically, the conversion factor currently used has considered building social function, geometric typology, slope & orientation, structural quality, economic cost, and shadow & obstacle.

Although the conversion factor currently used has been integrated for numerous aspects, it is a quantification of the average situation at the national level. City-level availability may vary, but due to the limitations of current data sources (e.g., policy-related data and three-dimensional data are difficult to obtain across more than 300 Chinese cities), we are unable to further refine the currently used factors to the city level.

However, in order to provide the readers with more information, we selected four representative cities with different administrative levels and implemented an alternative approach to clarify the differences in city-level rooftop availability based on the data sources that we can obtain. In our supplementary experiments, four major aspects, including building social function, geometric typology, slope & orientation, and shadow & obstacle, are considered.

Revision 2:

- Page 13, Line 5-13 and Page 14, Line 37-40: We have refined the description of accounting for available rooftop area, clarifying the aspects that have been considered for the currently used conversion factors.
- Supplementary Note 7: We have added the estimation process and results of the city-level rooftop availability conversion factor.

Comment 3:

How does the roof orientation etc. (solar suitability) compare with other countries. This might provide an argument for a more world-effort with rooftop installations. Could a table be added to partially summarize this?

Response 3:

Through supplementary studies, we give references of the differences in rooftop availability in different cities. However, due to high labor costs, we are unable to conduct similar studies at the urban scale in other countries for the time being.

Revision 3:

- Supplementary Note 7: We have added the estimation process and results of the city-level rooftop availability conversion factor.

Comment 4:

Is the built environment in general in China durable for this massive effort? The reason for asking is that in many countries, the most impact would be in the poorer areas (e.g., slums or favelas) where the support and security for PV (for example) may not be sufficient. Are there comments on this?

Response 4:

Thank you for raising this issue. Support and security are critical to promoting RSPV and tapping its technical potential. In fact, China has been committed to promoting PV for poverty alleviation and has undertaken great efforts to do so. China's Solar Energy Poverty Alleviation Program (SEPAP), for example, has been found to increase a county's disposable income per capita by about 7-8%. And the effectiveness is even better in poorer areas, especially for some counties in eastern China⁴. We have also added these findings to the discussion of economic feasibility to enrich this paper.

Revision 4:

- Page 13, Line 26-35: We have added China's support and security for the promotion of RSPVs to the discussion of economic feasibility.

Comment 5:

Are the cost comparisons between large plants and solar rooftops compelling?

Response 5:

Although a detailed economic assessment is beyond our scope, we have combined existing studies that have addressed this issue in detail to enrich our paper. Specifically, without subsidies, commercial and industrial DSPV in 100% Chinese cities has achieved user-side grid parity⁵, and household DSPV in 86% cities has been shown to be economically viable⁶.

Revision 5:

- Page 13, Line 26-35: We have added a discussion of the economic feasibility of RSPV deployment, providing references for its investment cost and levelized cost.
- Page 13, Line 26-35: We have discussed the profitability of household and commercial and industrial DSPV.

References

- 1 Jacobson, M. Z. *et al.* 100% Clean and Renewable Wind, Water, and Sunlight All-Sector Energy Roadmaps for 139 Countries of the World. *Joule* **1**, 108-121, doi:<https://doi.org/10.1016/j.joule.2017.07.005> (2017).
- 2 Zhong, T. *et al.* Assessment of solar photovoltaic potentials on urban noise barriers using street-view imagery. *Renewable energy* **168**, 181-194 (2021).
- 3 Zhang, K. *et al.* Quantifying the photovoltaic potential of highways in China. *Applied Energy* **324**, 119600 (2022).
- 4 Zhang, H. *et al.* Solar photovoltaic interventions have reduced rural poverty in China. *Nature communications* **11**, 1969 (2020).
- 5 Yan, J., Yang, Y., Elia Campana, P. & He, J. City-level analysis of subsidy-free

solar photovoltaic electricity price, profits and grid parity in China. *Nature Energy* **4**, 709-717 (2019).

- 6 Chen, H. & Chen, W. Status, trend, economic and environmental impacts of household solar photovoltaic development in China: Modelling from subnational perspective. *Applied Energy* **303**, 117616, doi:<https://doi.org/10.1016/j.apenergy.2021.117616> (2021).

Very sincerely yours,

Min Chen on behalf of the authors